# Effect of Additional Mass on Natural Frequencies of Weight-Sensing Structures

**DOI:** 10.3390/s23177585

**Published:** 2023-09-01

**Authors:** Guiyong Guo, Shuncong Zhong, Qiukun Zhang, Jianfeng Zhong, Dongming Liu

**Affiliations:** 1School of Mechanical Engineering and Automation, Fuzhou University, Fuzhou 350108, China; 13960922779@163.com (G.G.); qk_zhang@fzu.edu.cn (Q.Z.); zhongjianfeng@fzu.edu.cn (J.Z.); dongmingliu42@163.com (D.L.); 2Fujian Provincial Key Laboratory of Terahertz Functional Devices and Intelligent Sensing, School of Mechanical Engineering and Automation, Fuzhou University, Fuzhou 350108, China; 3Fujian Institute of Metrology, Fuzhou 350003, China; 4Fujian Key Laboratory of Force Measurement, Fuzhou 350100, China; 5Key Laboratory of Force Measurement for State Market Regulation, Fuzhou 350100, China

**Keywords:** weight-sensing structure, load cell, additional mass, natural frequency, resonance

## Abstract

The phenomena of variability and interference in the natural frequencies of weight-sensing structures applied in complex working conditions must solve the problem of reducing or eliminating resonance under low-frequency vibrations to maximize stability, accuracy and reliability. The influence laws of the additional mass with relevant characteristics on the natural frequencies, which include the components of mass, stiffness and center-of-mass distribution, etc. Firstly, the theoretical formulas of the mathematical model are given based on different characteristics of the weight-sensing structure, and various combinations of additional masses on the weight-sensing structures are adjusted in the X-, Y-, and Z-directions. The key factors to be specifically considered in the theoretical formulas are discussed through simulation analysis and experimental validation. Secondly, the locking strength of the fastening screws of some components was changed, and another component was placed on the experimental platform in the experiment. The results show that the mass, center-of-mass, stiffness distribution and other factors of the additional mass have different effects on the natural frequencies, which are important for the demand for high-precision, high-stability weighing measurement. The results of this research can provide an effective scientific evaluation basis for the reliable prediction of natural frequencies.

## 1. Introduction

The natural frequency of a mechanical system is a core parameter of reflecting dynamic characteristics, which plays an important role in many dynamic research fields such as vibration control, fault diagnosis, and structural optimization design [1,2,3,4,5,6]. Determining or calculating natural frequency is one of the main tasks of structural dynamic analysis, which has great theoretical value and practical engineering applications background [7]. With the development and application of weighing products and mechanical equipment, there is an increasing demand for high-precision and high-stability weighing measurements. The study of natural frequency has become an indispensable part of modern industrial production, quality control in high-end manufacturing and scientific research for optimal and precise measurements. Weighing measurement often requires the use of several weight-sensing structures, which are at the heart of the overall equipment. As a new type of weighing element, it is mainly composed of a load cell, load transfer mechanism, mounting connectors and instrumentation, etc., which can transform the mass signal into a measurable electrical signal and output it into the instrumentation for display. It is convenient to connect with various shapes of mechanical devices, such as hopper scales, electronic scales, mixing dynamics of weighing, truck-mounted garbage scales, etc., which can be divided into parallel beams, plates, columns, S-shapes, etc., according to their shapes. In this work, we focus on the most common parallel beam load cell with the resistance strain principle and the weight-sensing structure with the additional mass.

Most of the weight-sensing structures cannot exist independently, which may be affected by the vibration excitation of the surrounding environment in the actual measurement. The 10 Hz~1000 Hz range of low-frequency vibration sources are mostly caused by the inertia of the frame or shock vibration. It needs to identify and judge this phenomenon and take some suppression methods for improvement [7,8,9,10]. However, the factors affecting the natural frequencies of the weight-sensing structure consist mainly of the additional mass that is collectively constituted by the connecting mechanism other than the object to be weighed, the tooling, and other components on the base platform. It changes the natural frequencies to different degrees, which can easily cause the process wiring of the weight-sensing structure to be altered or even damaged, such as weld-off, loose connection, or even crack extension, etc., thus affecting the accuracy, reliability and stability of the overall output. Especially for some low stiffness, small mass and a small range weight-sensing structure, the additional mass greater than the body of the weight-sensing structure will cause a drastic change in the frequency, which in turn affects the dynamic performance. Therefore, the study of additional mass-related characteristics is important for the selection of suitable load cells and supporting tooling mechanisms for practical applications to ensure a sufficient range of available bands and obtain useful dynamic signals. By processing and analyzing the data, the causes of equipment failure and the prediction of possible equipment failures can be found, providing a scientific theoretical basis for the prevention of accidents and scientific arrangements of the overhaul.

In the practical application of force measurement or weighing, many scholars have conducted some related research. J. Vanwalleghem [11] et al. found that changing the experimental boundary conditions (e.g., load mass) affected the dynamic characteristics of the force transducer when performing calibration of rod strain force transducers. E. Korkmaz [12] observed a decrease in the natural frequencies of the force table due to an increase in the mass of the tooling in a cutting force application. Gu Baodong [13] et al. treated the resonance phenomenon caused by the decrease in the calibration frequencies of the force transducer after the addition of additional mass as “mass increase”. Xu [14] et al. used displacement measurements in collision experiments to study the effect of additional mass on the natural frequencies of the force-measuring wall. To summarize, these studies did not explore the quantitative relationship between the additional mass and the dynamic performance of the transducer in-depth, nor did they analyze the mechanism of action. Among the more systematic studies, the research system of Cha Fuyuan [15] and Yang Rui [16] adopted the step unloading method to study the multi-order natural frequencies change of strain transducers under five different kinds of additional mass, but the dynamic calibration “loading head” was too large for its mass, which led to the calibration of the blind spot in the study of low additional mass. 

Yang Yiliu [17] analyzed the factors affecting the resonance frequency of MEMS silicon micro-cantilever beams with a double bridge-arm structure. However, only theoretical and simulation analyses were conducted from the perspective of the increasing of additional mass and the increasing of the proportion of the mass block length to the total length of the micro-cantilever beam, and no experimental validation was carried out. Although Zhang Can [18] studied the quantitative relationship between the additional mass and the dynamic performance of the sensor but did not quantitatively analyze the influence of the additional mass structure, the center-of-mass position and the connecting parts of the tightness of the trend of change, so its research has certain limitations. In this work, through theoretical derivation, simulation analysis and experimental validation of the research method, the different relevant characteristics of the mass distribution of the additional mass, the center-of-mass distribution, the stiffness distribution, etc., were revealed to influence the natural frequency law of the weight-sensing structure. The excitation frequency is measured according to the composition of the surrounding environmental conditions and the available frequency range of the weight-sensing structure is calculated to reduce or eliminate the resonance under the low-frequency vibration excitation in the surrounding environment. Ultimately, it provides a design basis for improving the stability, accuracy and reliability of the weight-sensing structure.

## 2. Principle and Theoretical Model

Figure 1 depicts the schematic diagram of the weight-sensing structure, which is mainly composed of a parallel beam load cell and the additional mass and other parts. The mass distribution, stiffness distribution and center-of-mass distribution of the additional mass are the focus of this work.

The additional mass generally refers to secondary loads such as tools, actuators, fixtures, etc., in addition to the table surface, which is characterized by high mass, low stiffness and complexity compared to the preloaded components. Therefore, in practice, it is necessary to consider the influence of the structural parameters of the secondary loads on the relevant characteristics of the existing weight-sensing structure. Where $L_{1}$, B and $H_{1}$ is the length, width and height of the load cell elastomer in the weight-sensing structure, respectively, are symmetrically distributed in the center-line of the horizontal direction of the additional mass and the external dimensions of the weight-sensing structure are shown in Table 1.

### 2.1. Theoretical Model

The relationship between the relevant characteristics of the additional mass and the natural frequencies of the weight-sensing structure is the object of study in this work. Therefore, in practical applications, the frequency-domain index of the weight-sensing structure is generally used as a standard to measure its dynamic characteristics. Under a specific external excitation frequency, the resonance of a weight-sensing structure appears at the frequency when a large natural vibration occurs. To determine the available frequency range, it is necessary to study the fundamental laws of the natural frequencies of a weight-sensing structure with an additional mass to solve the unpredictable superposition effect on the accuracy, which has a direct impact on the stability, accuracy and reliability of the weight-sensing structure. Generally, the weight-sensing structure is fixed at one end to a rigid surface such as a base or foundation, and its dynamic characteristics can be simplified, as shown in Figure 2, a simple “mass-spring-damper” of the system. The weight-sensing structure consists of a fixed base, a parallel beam load cell and several additional masses at its free end, where ∆m is the quality of the preloaded structure on the right side of the load cell in the weight-sensing structure and is the equivalent damping.

To obtain the natural frequencies of the weight-sensing structure with the additional mass, it is necessary to make the following assumptions: the additional mass preloaded structure is connected to the load cell, and the connection of the load cell to the mounting base is rigid. Then, the weight-sensing structure will produce a displacement of the right additional mass preloaded structure under the action of external force. The differential equations for the equilibrium state of the system are as follows:(1)$$\left(m_{0}+∆m\right)\cdot \frac{d^{2}y\left(t\right)}{dt^{2}}+(c+∆c)\cdot \frac{dy\left(t\right)}{dt}+\left(k+\Delta k\right)\cdot y(t)=x(t)$$

The Laplace transformation of Equation (1) is given as follows:(2)$$H\left(s\right)=\frac{\omega_{n}^{2}\cdot K}{s^{2}+2\xi \cdot \omega_{n}\cdot s+\omega_{n}^{2}}$$
where K is the sensitivity of the system, and $K=1/\left(k+\Delta k\right)$, set K= 1 for ease of calculation; $\omega_{n}$ is the natural angular frequencies of the system, $\omega_{n}=\sqrt{\frac{k+\Delta k}{m_{0}+∆m}}$, ξ is the damping ratio of the system, $\xi =\frac{c+∆c}{2\sqrt{\left(m_{0}+m\right)\cdot \left(k+\Delta k\right)}}$.

Substituting s=j·ω into Equation (2), the expression for the amplitude–frequency characteristic of the weight-sensing structure can be obtained as follows:(3)$$A\left(\omega \right)=\frac{1}{\sqrt{{\left[1-{\left(\frac{\omega }{\omega_{n}}\right)}^{2}\right]}^{2}+{\left[2\xi \left(\frac{\omega }{\omega_{n}}\right)\right]}^{2}}}$$

The dynamic characteristics of a weight-sensing structure can be characterized in the time and frequency-domains, both of which can be interchanged through the Fourier transform, but the frequency-domain characterization is often more concise and easier to understand. In practical applications, the frequency domain is more often used as a measure of the dynamic characteristics of the weight-sensing structure. The natural frequencies $f_{n}$ of the weight-sensing structure are converted from the angular frequencies $\omega_{n}$, thus,
(4)$$f_{n}=\frac{1}{2\pi }\omega_{n}=\frac{1}{2\pi }\sqrt{\frac{k+\Delta k}{m_{0}+∆m}}$$

From Equation (4), it can be seen that the natural frequencies of the weight-sensing structure are mainly influenced by the distribution of the structural stiffness and the distribution of the mass. The mass includes the mass of the weight-sensing structure itself, as well as the additional mass that inevitably needs to be added to connect the end of the actuator. The structure, mass and distribution of this additional mass in space are unpredictable and depend mainly on the purpose and function of the weight-sensing structure and the equipment itself. Since the additional mass can be measured in advance, the natural frequencies of the weight-sensing structure can be obtained here by simply asking for k+Δk. According to the classical theory of beam bending in the mechanics of materials, the flexural differential equation of the load cell in the weight-sensing structure is as follows:(5)$$\frac{d^{2}z}{d^{2}x}=\frac{F}{EI}\left(L_{1}+L_{2}-x\right)$$
where F is the concentrated load applied to the T-shaped loading pad at the free end connected to the load cell, which is along the Z-direction, E is the modulus of elasticity of the load cell and the material of the additional mass; I is the moment of inertia of the micro-cantilever beam. It is divided into two regions for solving. According to the different components, the load cell is considered the first region, i.e., $0<{x\ll L}_{1}$, and the T-shaped loading pad at the free end (the first additional mass) is considered as the second region, i.e., $L_{1}<{x\ll L}_{2}$, from which Equation (5) can be expressed as follows:(6)$$\frac{d^{2}z}{d^{2}x}=\left\{\begin{array}{c} \frac{d^{2}z_{1}}{d^{2}x}=\frac{F}{EI_{1}}\left(L_{1}+L_{2}-x\right);\;0<{x\ll L}_{1}\\ \frac{d^{2}z_{2}}{d^{2}x}=\frac{F}{EI_{2}}\left(L_{1}+L_{2}-x\right);\;L_{1}<{x\ll L}_{2} \end{array}\right.$$
where $I_{1}$ and $I_{2}$ in Equation (6) are the rotational inertia of the load cell and the T-shaped loading pad (the first additional mass), respectively, of the two parts of the equal cross-section after splitting. Based on the above theoretical formulas, the factors affecting the natural frequencies of the weight-sensing structure can already be obtained, which can be used as a reference for the following in-depth simulation analysis and experimental validation and will not be repeated here. The solid structure of the weight-sensing structure is simplified into the model in Figure 3, where the mass of the load cell is $m_{0}$ and the mass of the additional mass is ∆m, $∆m=m_{1}+m_{2}+\ldots +m_{i}\;(i$ = 1, 2, …, n).

To analyze the laws of influence on the dynamic characteristics of the weight-sensing structure with additional mass, a 3-D solid model needs to be established. The 3-D solid model includes a type of additional mechanism based on a relatively uniform load distribution (e.g., for high-precision electronic scales with a large number of applications, where the additional mass corresponds to the steel support frame). The mass of the additional mechanism is generally not too large (e.g., up to 5 kg) and the load distributions in the X-, Y-, and Z-directions are relatively homogeneous so that there is hardly any concentration of loads in certain areas. The additional mass is interconnected with the weight-sensing structure, which causes the mass to change and the natural frequencies to change accordingly. This means that the operational accuracy and stability in practical applications will be limited by the additional mass if no suppression measures are taken. According to the theoretical model above, the additional mass will change the mass distribution of the weight-sensing structure, thus affecting the natural frequencies and damping ratio. Therefore, the additional mass is also a cause of resonance in the weight-sensing structure.

### 2.2. Preliminary Simulation Analysis

The above theoretical modeling equations show that the natural frequencies of a weight-sensing structure are dependent on mass and stiffness. However, since the weight-sensing structure is not perfectly symmetrical in the X-, Y-, and Z-directions, and the actual additional mass varies in shapes. It requires detailed analytical research on the relevant characteristics of the additional mass. As shown in Figure 4 for the finite element model of the weight-sensing structure, the ABAQUS finite element software is used to simulate and analyze the effect of the unilateral X-direction additional mass on the first and second-order natural frequencies, which includes the mass and center-of-mass change, as well as the mass and stiffness change, to obtain the preliminary law that provides a reference for further in-depth study of simulation and experimental validation.

The weight-sensing structure model is simplified appropriately. Its grid size is 2~4 mm, the cell grid type is an eight-node linear hexahedral cell, and the number of grids is 78,962. The load cell is made of 2A12 aluminum alloy with a yield strength of 340 MPa, tensile strength of 485 MPa, modulus of elasticity of 70 GPa, Poisson’s ratio of 0.3, and density of 2800 $kg/m^{3}$. The rest of the components are made of 40Cr steel, with a yield strength of 785 MPa, tensile strength of 980 MPa, elastic modulus of 210 GPa, Poisson’s ratio of 0.3, and density of 7800 $kg/m^{3}$. Among them, the base of the weight-sensing structure weighs about 3.70 kg, the load cell weighs about 3.50 kg, the T-shape loading block (the 1st additional mass) weighs about 2.40 kg, and the rest of the additional mass weighs about 1.5 kg. The rest of the additional mass weighs about 1.75 kg. After modeling, a frequency analysis step is created and the Lanczos eigenvalue solver is used to obtain the first two orders of the natural frequencies of the weight-sensing structure. Figure 5 shows the influence of the additional mass on the “mass, center-of-mass, natural frequencies” and “mass, stiffness, natural frequencies” for the unilateral single-row extension of the mass in the X-direction, respectively.

The simulation analysis results in Figure 5 above reveal that the additional mass distribution has a greater effect on the natural frequencies. The larger the mass, the lower the natural frequencies. In addition, the stiffness distribution of the additional mass is also an important factor affecting the natural frequencies; the greater the stiffness, the greater the natural frequencies. However, the influence of mass distribution and stiffness distribution in the X-, Y-, and Z-directions need further in-depth simulation analyses and experimental validation, which is a key factor to be considered in the optimization design of the weight-sensing structure in practical applications, especially in the high-accuracy applications.

## 3. Detailed Simulation Analysis and Experimental Validation

The hammering method is a common solution to the measurement of the natural frequency of solid components [19]. Several researchers have conducted studies on the subject, and the final experimental results were favorable. Li Yingjun [20] conducted a dynamic analysis of a six-dimension-force piezoelectric test system for large, load-bearing manipulators and obtained the natural frequency of the test system by using the hammering method. Meng Xie [21] conducted a study on the clip’s natural frequencies of a DT-III fastener system with and without a train load and were compared and analyzed to study the respective effects of the above two factors, which were verified by the hammer test above.

### 3.1. Experimental Design

To further validate the correctness of the analytical laws of simulation, the additional mass of the weight-sensing structure is experimentally tested in various combinations. The experiment focuses on the different working conditions from the mass distribution, center-of-mass distribution, and stiffness distribution of the additional mass, which causes the change of the natural frequency characteristics of the weight-sensing structure. The experiment is divided into “mass distribution, natural frequencies”, “center-of-mass distribution, natural frequencies”, “stiffness distribution, natural frequencies”, “other components, natural frequencies,” and “fastening screws, natural frequencies” are analyzed and validated.

As shown in Figure 6, the weight-sensing structure is placed on a thick pad and locked with screws and then installed on a rigid, large-sized steel experimental platform with no other parts. Then, two accelerometers are pasted near the strain gauges of the load cell elastomer in the weight-sensing structure, and a steel force hammer with a semicircular head and a diameter of 200 g is used as an excitation tool. At the same time, the hammer and the two accelerometers are respectively connected to the dynamic test and analysis system and the computer so that the force hammer strikes the strain gauges of the weight-sensing structure under the test in the area near the strain gauges and the bearing part, thus generating a stable frequency response output. Then, MATLAB was used to generate the natural frequency spectrogram of the weight-sensing structure through the two accelerometers. The force hammer spectrogram was obtained by selecting three groups of single strike data obtained by the force hammer excitation spectrogram, as well as the two accelerometers at the same time. The acquisition of the spectrogram of the first wave peaked and the second wave peaked as the first- and second-order natural frequencies, and the three groups of data took the average value of the measured values.

### 3.2. Experimental Results

#### 3.2.1. Effect of Mass Distribution on the Natural Frequencies

This paper considers different loads not based on strict criteria but on the coordination of incremental loads. The other considerations are as follows: (1) Two-ton load cells made of aluminum alloy have a large number of applications, and manufacturers need to focus on improving the core components, which weigh about 3.50 kg and have common structural attributes. (2) The T-shaped loading pads (the first additional mass) weigh about 2.40 kg, which is mainly designed according to the basic structure of the maximum load capacity for the 2 tons. (3) The rest of the additional mass involves a variety of ways to extend the combination and tends to take half of the load cell mass weighing about 1.75 kg for design, but also takes into account the quality of the more intuitive. In fact, for some high-precision electronic scales of the steel support frame, its load is a random weight, and its mass even has a certain degree of randomness, so this paper’s research focuses on the weight-sensing structure with the additional mass of the sequential increase in the change rule of the actual natural frequency.

To examine the effect of the additional mass on the natural frequencies of the weight-sensing structure, the effect of the additional mass in the X-, Y-, and Z-directions on the first- and second-order natural frequencies of the weight-sensing structure is simulated and analyzed, and it is explored whether the combination of the mass distributions causes the natural frequencies of the weight-sensing structure to undergo a more significant change, and then experimental validation is carried out, to reveal the effect of the mass distributions on the natural frequencies of the law. Table 2 lists effect of mass distribution on the natural frequencies. The influence of X additional mass (single-row length extension X-1, double rows length extension X-2 and thickness extension X-3) on the first- and second-order natural frequencies of the weight-sensing structure is studied, and the results of simulation analysis and experimental validation are as follows:

In accordance with the above method, it can be similarly derived from the additional mass in the Y- and Z-directions, such as the weight-sensing structure of the first and second-order natural frequency results. Because its rule of law is very similar and the space limitations of its detailed data will not be repeated here, Figure 7 shows the corresponding curves.

As shown in Figure 7 above, from the simulation analysis and experimental validation of the above the X-, Y-, and Z-directions, the natural frequencies of the weight-sensing structure decrease sharply with the increasing of the additional mass up to m = 2.4 kg; when the additional mass m > 2.4 kg, the decreasing of the natural frequency tends to be flat; when the additional mass m > 9.4 kg, the natural frequencies is as low as 50 Hz, which is gradually approaching to the low-frequency phenomenon. Based on the fact that the load cell in the weight-sensing structure weighs about m = 3.50 kg and the T-shaped loading pad (The first additional mass) weighs about m = 2.40 kg, the difference in mass size between the two is not significant, and according to the theoretical model of Equation (4), one of the reasons for the sharp decline lies in the size of the mass multiplier between them, and the natural frequencies will decrease accordingly with the growth of the additional mass according to the trend, and the natural frequencies will be reduced by the external resonance and low frequency of resonance. The probability of low-frequency interference from external resonance is increased, resulting in a greater impact on the measurement accuracy of the weight-sensing structure.

Table 3 lists the measured values of the different additional mass when the length of a single row is extended in the X-, Y-, and Z-directions. Comparing the variability and influence on the natural frequencies, the rest of the combinations of the measured values will not be repeated. However, a variety of additional mass in the weight-sensing structure will be given, as shown in Figure 8. The measured values of the remaining combinations will not be repeated, but a comparison of the differential effects of additional mass on the natural frequencies in the X-, Y-, and Z-directions will be given.

Through the above simulation and experimental validation, regardless of the first- and second-order natural frequencies, the simulation analysis and experimental validation have some differences. The main reason for these differences is that the simulation model is simplified, and in comparison, the experimental platform has some idealization in the X-, Y-, and Z-directions of the weight-sensing structure. The growth in the additional mass of the natural frequencies is a decreasing trend is the same between the simulation analysis and experimental validation, and the size of the natural frequencies of the law for the Y- and Z-directions, is smaller than the X-direction. The size of the natural frequencies in the Y- and Z-directions is smaller than that in the X-direction, because the X-direction is perpendicular to the main load-bearing direction of the weight-sensing structure, which is more sensitive to the effect of the natural frequencies with the increase of the additional mass. The trend of the law shows that in the weight-sensing structure, the design of additional institutions needs to consider the three-dimensional space on the structural optimization and arrangement, especially in terms of the X-direction of the relevant characteristics of the impact of the weight size. The three directions of the variability of the law of change are different and not linear and also reveal that the measured object of online measurement and long-term monitoring must be a focus to align with the actual application of the working conditions of the band range of the natural frequencies, and also as a resonance or resonance inhibition measures in engineering applications of the scientific basis.

#### 3.2.2. Effect of Center-of-Mass Distribution on the Natural Frequencies

Three different center-of-mass distributions of the additional mass are studied to simulate and analyze the effects of the first- and second-order natural frequencies of the weight-sensing structure and experimentally validate them. Furthermore, these distributions explore whether the center-of-mass distribution of the additional mass causes a more significant change in the natural frequencies of the weight-sensing structure and thus reveal the influence of the center-of-mass distribution on the natural frequencies. As shown in Figure 9 and Figure 10, the following center-of-mass offset distances according to the unilateral X, Y, and Z-directions of the various extension combinations of order, and in the extension of the combinations can be obtained under the same additional mass of 3 (m  = 5.9 kg) at this point for the center-of-mass position of different comparisons, the additional mass of 5 (m = 9.4 kg) when the same way of processing, respectively, the following simulation analysis and experimental validation.

As shown in Figure 11, the results of simulation analysis and experimental validation under each center-of-mass distribution are as follows:

From the above results, the natural frequencies show an overall decreasing trend with the increase of the center-of-mass offset distance, which is close to the inversely proportional linear monotonically decreasing relationship, and the decreasing slopes in the X-, Y-, and Z-directions are not very much different from each other. It can be seen that the spatial distribution of the center-of-mass of the additional mass of the weight-sensing structure is related to the natural frequencies; in general, the closer the center-of-mass of the additional mass is to the center, the higher its natural frequencies. Therefore, the center-of-mass distribution should be designed and evaluated as an important index that should be used to assist in obtaining the variable natural frequency range of the weight-sensing structure and the equipment under the actual application conditions; these can be used as an important basis for the resonance suppression measures in engineering applications.

#### 3.2.3. Effect of Stiffness Distribution on the Natural Frequencies

As shown in Figure 12 and Figure 13, three different stiffness distributions of the same additional mass are studied to simulate and analyze the effects of the first- and second-order natural frequencies of the weight-sensing structure and experimentally validate them to explore whether the stiffness distributions of the additional masses cause the natural frequencies of the weight-sensing structure to change significantly, and thus to reveal the effects of the stiffness distributions on the natural frequencies. The following stiffness distribution is also according to the unilateral X-, Y-, and Z-directions of the various extension combinations of the order, and the extension of the combination can be obtained under the same additional mass for 3 (m = 5.9 kg) at this point of time the size of the stiffness of the different comparisons, the additional mass of 5 (m = 9.4 kg) for the same way to deal with the additional mass of 5 (m = 9.4 kg) when the same way to deal with the additional mass of 5 (m = 9.4 kg) and to intuitively show the trend of the laws of the X-, Y-, and Z-directions, the scale of the transverse and longitudinal coordinate to show the trend of the X-, Y-, and Z-directions, the scales of horizontal and vertical coordinates are kept the same.

As shown in Figure 14, the results of simulation analysis and experimental validation under different stiffness distributions are as follows:

From the above results, the simulation analysis and experimental validation of the natural frequencies with the increase of the stiffness is an upward trend; the trend is the same and close to the linear monotonic incremental relationship. It can be seen that the natural frequencies of the additional mass of the weight-sensing structure also depend on the stiffness distribution, and according to the slope of the X-direction is the largest, Y-direction is the second largest, and Z-direction is the most gentle, the main reason for this phenomenon is related to the use of the weight-sensing structure when the main state of force is Z-direction. In addition, the stiffness distribution is closely related to the structural characteristics, and should be considered in the structure of the additional mass of the weight-sensing structure, which is conducive to the effective control of the natural frequencies in a certain band range, obtaining the variable natural frequency range in line with the actual application of the working conditions, and also serves as an important basis for the resonance suppression measures in the engineering applications.

#### 3.2.4. Effect of Other Components Placed on the Experimental Platform on the Natural Frequencies

To study the effect of other components placed on the experimental platform on the natural frequencies of the weight-sensing structure, as shown in Figure 15, a speed reducer weighing about 100 kg is randomly placed on the experimental platform at a distance of 1000 mm and 100 mm near the location of the natural frequencies of the weight-sensing structure at this point in time to measure the natural frequencies of the weight-sensing structure, to determine the degree of influence on the natural frequencies of qualitatively.

From the results of Figure 15 above, the variation in the natural frequency waveform graphs captured by the two accelerometers is not significant, and the differences between their first-order and second-order natural frequencies are only 0.06% and 0.08%, respectively, which are negligible to some extent. The main reason for this phenomenon is that the connection between the platform and the laboratory foundation is sufficiently solid, and the rigid connection between the weight-sensing structure and the platform and the pads is better. These conditions are sufficient to suppress the influence of parasitic interference effects introduced by other components on the platform.

#### 3.2.5. Effect of Loose Fastening Screws on the Natural Frequencies of Weight-Sensing Structure

As the weight-sensing structure grows in service life, or as resonance occurs due to external forces, or even due to changes in the surrounding environment, the fastening screw connections in the weight-sensing structure may loosen as a result. To qualitatively assess whether the loosening of the fastening screws before and after causes a large change in the natural frequencies, which in turn provides an assessment basis for the later online monitoring of in-use weight-sensing structure and equipment. As shown in Figure 16, previously, the load cell and the base between the use of four strengths of 12.9-grade M16 hexagon socket head-fastening screws for connection, and each screw that locks can be applied to the maximum torque of about 150 Nm. This experiment loosened the four screws so that the applied torque was reduced to about 30 Nm; the other conditions remained unchanged, as well as the weight at this time. The natural frequencies of the weight-sensing structure were measured.

From the results of Figure 16 above, the waveform diagrams of the natural frequencies acquired by the two accelerometers significantly changed, and the variability of their first-order and second-order natural frequencies is as high as 79.8% and 43.3% compared with that before the tightening screws were loosened. This phenomenon will cause a misjudgment of the available frequency range of the natural frequencies, and resonance may occur without being detected. It indicates that the tightness of the main fastening screws of the weight-sensing structure is extremely important and has a significant influence on the accurate determination and evaluation of the natural frequencies.

## 4. Conclusions

This work analyzes the phenomena of variability and interference in the natural frequencies of a weight-sensing structure applied in complex working conditions to solve the problem of reducing or eliminating the resonance interference under the excitation of low-frequency vibration. In this work, we analyzed the relevant characteristics of the additional mass and the laws of the additional mass with relevant characteristics on the natural frequencies, which include the components of mass, stiffness and center-of-mass distribution, etc. Based on the characteristics of mass, stiffness and other variables in the calculation formula of the theoretical model, we adjusted the additional mass of the weight-sensing structure of the single-row length extension, double-row length extension, thickness extension and other combinations. Finite element simulation analysis and experimental validation were used; in addition, the locking strength of the fastening screws of some components and the distribution conditions of the ancillary components around the experimental platform were also changed. The conclusions could be drawn as follows:(1)Weight sensing structures with the growth of additional mass in the X-, Y-, and Z-directions: The natural frequencies first showed a sharp decline and then tended to level off. They can be as low as 50 Hz below the low-frequency phenomenon, the three directions of the downward trends are the same, and the size of the natural frequencies in the Y- and Z-directions are smaller than the X-direction; it is necessary to consider the three-dimensional direction of the arrangement of the problem.(2)The natural frequencies of the weight-sensing structure have a linear, monotonically decreasing relationship with the center-of-mass offset distance, and the natural frequencies are higher when the center-of-mass is closer to the initial center position, while the decreasing slopes in the X-, Y-, and Z-directions do not show a significant difference.(3)When the additional mass is constant, the natural frequencies of the weight-sensing structure have a linear, monotonically increasing relationship with the stiffness. The slope is largest in the X-direction, second largest in the Y-direction, and the flattest in the Z-direction.(4)The influence of other components on the platform on the natural frequency is minimal. The parasitic effects introduced by other components can be effectively suppressed when the platform or the weight-sensing structure is mounted securely enough.(5)As the service life of the weight-sensing structure grows or by external forces, some fastening screw connections could loosen, and the variability of the first- and second-order natural frequencies is as high as 79.8% and 43.3%, which may result in a misjudgment of the frequency band range.

There are several discrepancies between the simulation analysis and experimental validation results of the natural frequencies of the weight-sensing structure. The errors mainly come from the model used for simulation analysis, which has been simplified, and the physical object and the connection with the experimental platform is difficult to reach the ideal installation state. However, because of these variations, the natural frequencies of the weight-sensing structure in engineering applications should be measured online and monitored over time to obtain a range of available frequency bands.

To further improve the stability, accuracy and reliability of the weight-sensing structure, the following issues will be further studied in the future. Based on the influence of additional mass-related characteristics, a method of evaluating the natural frequencies of the weight-sensing structure with the weighted coefficients of variable factors should be proposed. These are used to solve the problems of measurement, diagnosis, prediction and suppression of external vibration interference of the natural frequencies under complex working conditions in industrial applications. Ultimately, it ensures that the weight-sensing structure maintains a good performance for a long time.

## Figures and Tables

**Figure 1 sensors-23-07585-f001:**
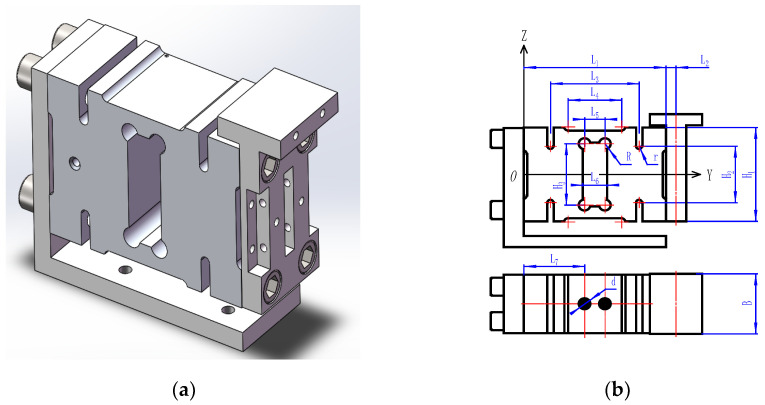
The model of the weight-sensing structure (including a T-shaped pad): (**a**) 3-D solid model of the weight-sensing structure; (**b**) the external dimensions of the weight-sensing structure.

**Figure 2 sensors-23-07585-f002:**
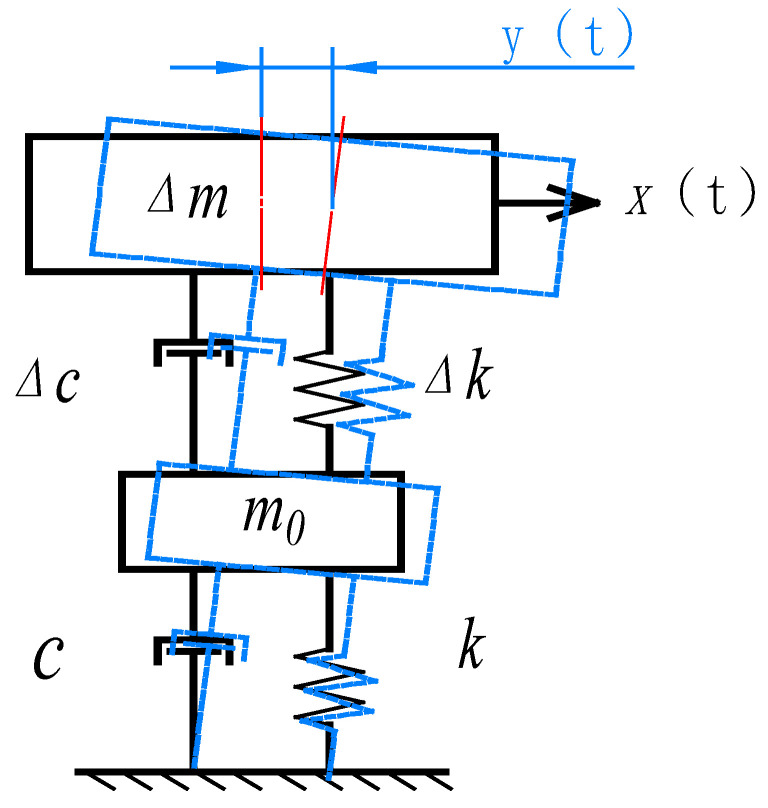
Weight-sensing structure with the simplified mass-spring-damper model.

**Figure 3 sensors-23-07585-f003:**
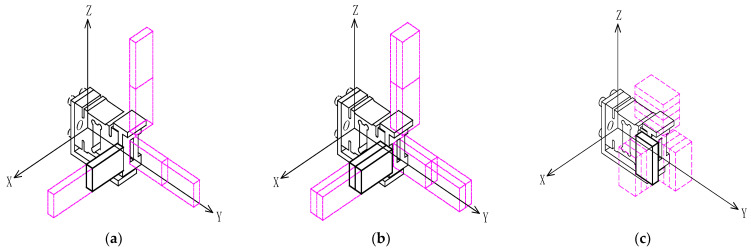
Combined extensions of the additional mass of the weight-sensing structure in the X-, Y-, and Z-directions: (**a**) The diagram of single-row length extension combination, named distribution form of X-1, Y-1, Z-1; (**b**) The diagram of double-rows’ length extension combination, named distribution form of X-2, Y-2, Z-2; (**c**) The diagram of thickness extension combination, named distribution form of X-3, Y-3, Z-3.

**Figure 4 sensors-23-07585-f004:**
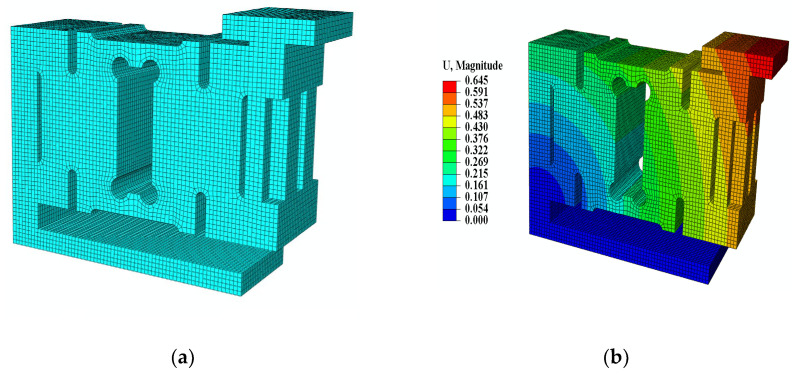
Finite element model of the weight-sensing structure: (**a**) Finite element meshing diagram; (**b**) Finite element cloud diagram (amplitude).

**Figure 5 sensors-23-07585-f005:**
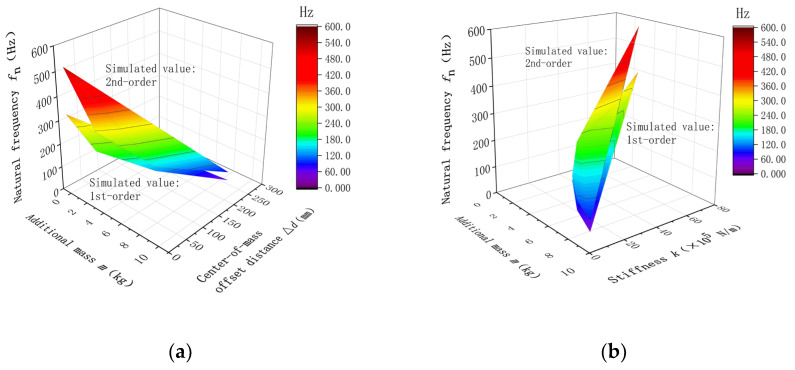
Influence of natural frequencies (an example in the X-direction): (**a**) The relationship of “mass, center-of-mass, and natural frequencies”; (**b**) The relationship of “mass, stiffness, and natural frequencies”.

**Figure 6 sensors-23-07585-f006:**
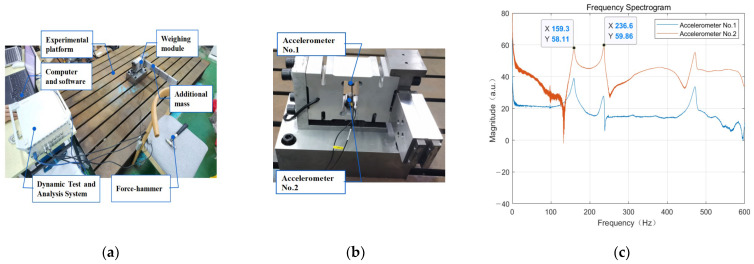
The experimental validation site of the weight-sensing structure’s natural frequencies: (**a**) General view of the experimental site; (**b**) The mounting positions of the two accelerometers; (**c**) One of the spectrograms in the experiment validation.

**Figure 7 sensors-23-07585-f007:**
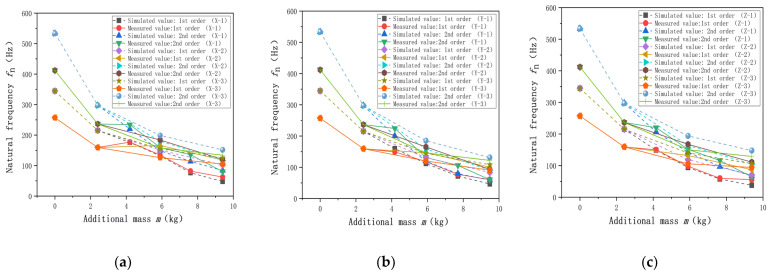
Distribution law of the natural frequencies with additional mass in the X-, Y-, and Z-directions: (**a**) The law in the X-direction; (**b**) The law in the Y-direction; (**c**) The law in the Z-direction.

**Figure 8 sensors-23-07585-f008:**
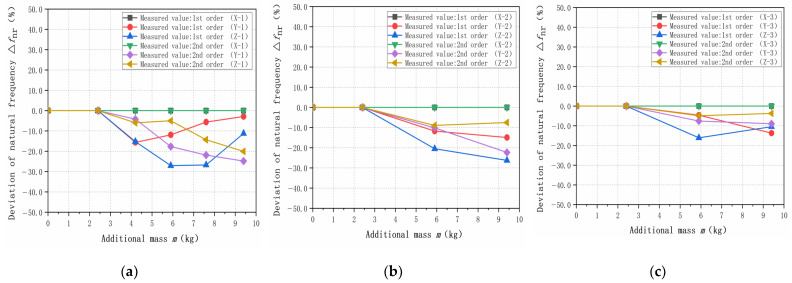
Comparison of the differential effects of additional mass on the natural frequencies in the X-, Y-, and Z-directions: (**a**) The law in the X-direction; (**b**) The law in the Y-direction; (**c**) The law in the Z-direction.

**Figure 9 sensors-23-07585-f009:**
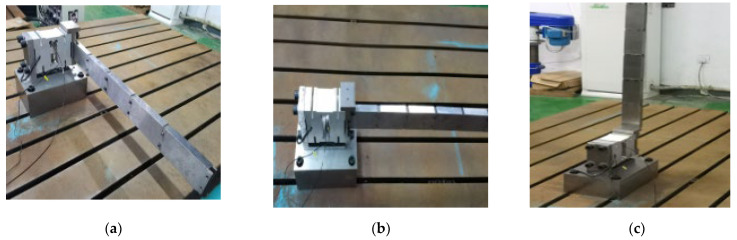
The photograph of center-of-mass distribution with single-row additional mass in the X-, Y-, and Z-directions: (**a**) The additional mass in the X-direction; (**b**) The additional mass in the Y-direction; (**c**) The additional mass in the Z-direction.

**Figure 10 sensors-23-07585-f010:**
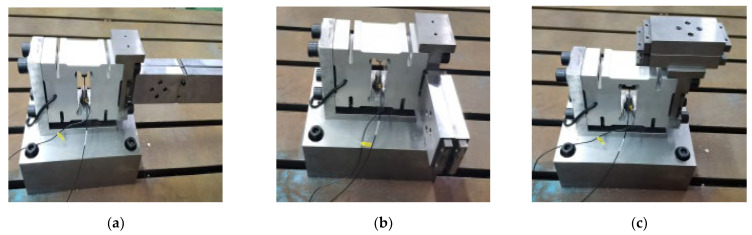
The photograph of center-of-mass distribution with double rows’ additional mass in the X-, Y-, and Z-directions: (**a**) The additional mass in the X-direction; (**b**) The additional mass in the Y-direction; (**c**) The additional mass in the Z-direction.

**Figure 11 sensors-23-07585-f011:**
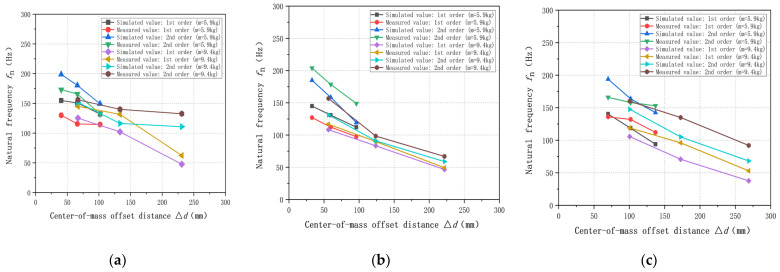
Effect of center-of-mass distribution on the natural frequencies in the X-, Y-, and Z-directions: (**a**) The law in the X-direction; (**b**) The law in the Y-direction; (**c**) The law in the Z-direction.

**Figure 12 sensors-23-07585-f012:**
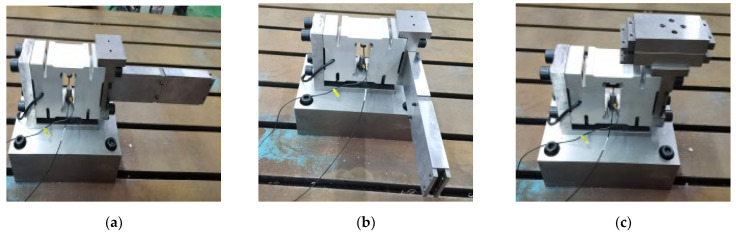
The photograph of stiffness distribution with single-row additional mass in the X-, Y-, and Z-directions: (**a**) The additional mass in the X-direction; (**b**) The additional mass in the Y-direction; (**c**) The additional mass in the Z-direction.

**Figure 13 sensors-23-07585-f013:**
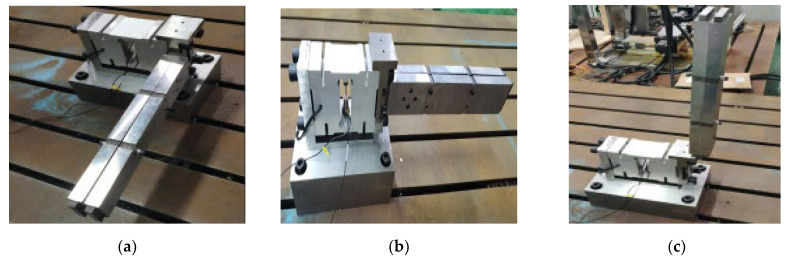
The photograph of stiffness distribution with double rows’ additional mass in the X-, Y-, and Z-directions: (**a**) The additional mass in the X-direction; (**b**) The additional mass in the Y-direction; (**c**) The additional mass in the Z-direction.

**Figure 14 sensors-23-07585-f014:**
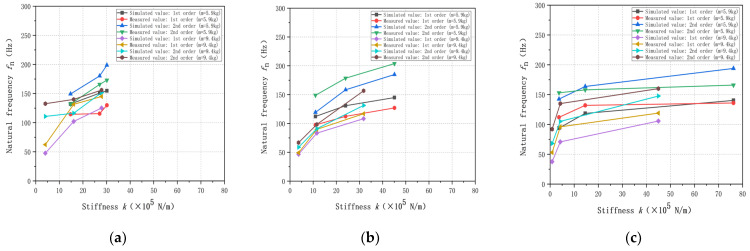
Effect of stiffness distribution of additional mass on the natural frequencies in the X-, Y-, and Z-directions: (**a**) The law in the X-direction; (**b**) The law in the Y-direction; (**c**) The law in the Z-direction.

**Figure 15 sensors-23-07585-f015:**
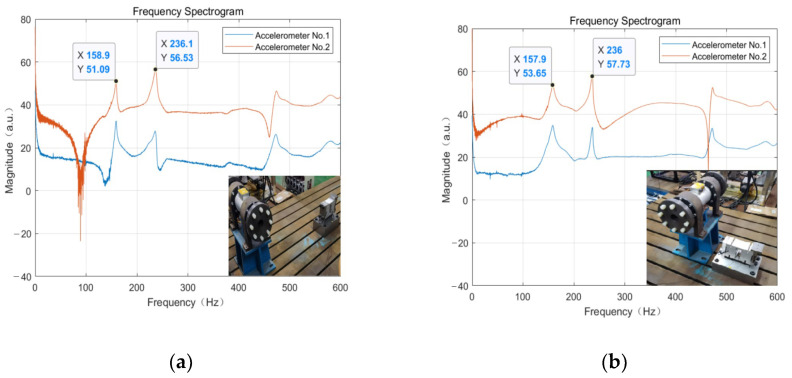
The spectrogram of the weight-sensing structure when other components are placed on the platform: (**a**) Far away from the platform; (**b**) Close to the platform.

**Figure 16 sensors-23-07585-f016:**
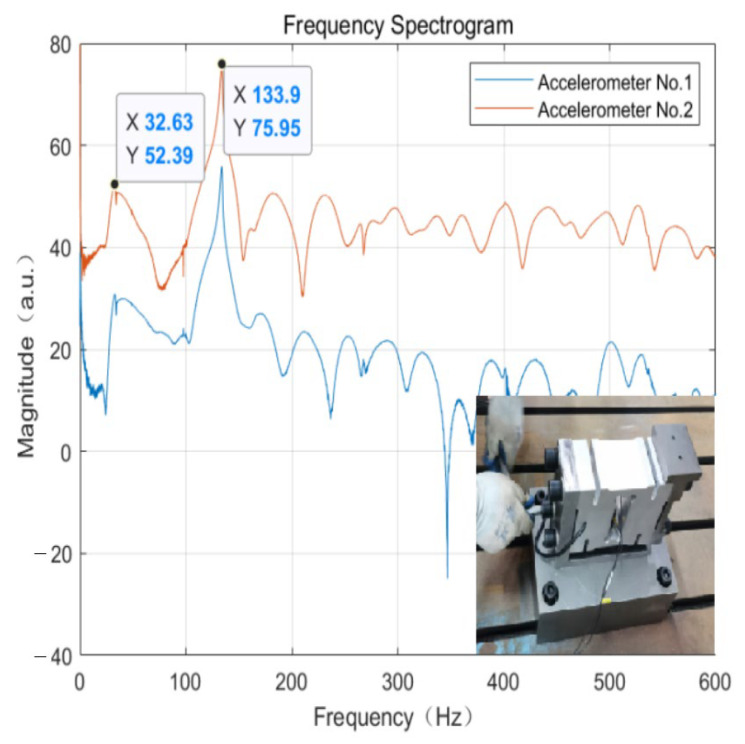
The spectrogram of the weight-sensing structure (after loosening the fastening screws).

**Table 1 sensors-23-07585-t001:** External dimensions of the weight-sensing structure.

Parameters	Symbol	Numerical Value (mm)
Total length of elastomer	$L_{1}$	176
Distance between the center-line of the T-shaped pad at the free end and the end face of the elastomer	$L_{2}$	9
Length of the center of the rounded corner at the bottom of the narrow groove	$L_{3}$	110
Length of strain gauge patch area	$L_{4}$	66.2
Center length of hollow groove fillet	$L_{5}$	25
Hollow slot width of strain gauge patch area	$L_{6}$	30
Total height of elastomer	$H_{1}$	125
Center width of the rounded corner at the bottom of the narrow slot	$H_{2}$	75
Center width of hollow groove fillet	$H_{3}$	81.7
Total width of elastomer	B	80
Radius of hollow groove in patch area	R	7.5
Bottom fillet radius of narrow groove	r	4

**Table 2 sensors-23-07585-t002:** Effect of mass distribution on the natural frequencies (an example in the X-direction).

Additional Mass (kg)	Distribution Forms	1st-Order Natural Frequency			2nd-Order Natural Frequency		
		Simulated Value (Hz)	Measured Value (Hz)	Deviation (%)	Simulated Value (Hz)	Measured Value (Hz)	Deviation (%)
0	X-1	344.4	257.0	−25.4	533.4	412.0	−22.8
	X-2	344.4	257.0	−25.4	533.4	412.0	−22.8
	X-3	344.4	257.0	−25.4	533.4	412.0	−22.8
2.4	X-1	215.1	159.3	−25.9	297.5	236.5	−20.5
	X-2	215.1	159.3	−25.9	297.5	236.5	−20.5
	X-3	215.1	159.3	−25.9	297.5	236.5	−20.5
4.2	X-1	176.1	177.0	+0.5	220.2	234.0	+6.3
	X-2	/	/	/	/	/	/
	X-3	/	/	/	/	/	/
5.9	X-1	132.1	134.4	+1.7	149.3	159.5	+6.8
	X-2	150.8	166.0	+10.1	180.3	183.8	+1.9
	X-3	154.8	126.0	−18.6	198.7	158.0	−20.5
7.6	X-1	76.4	81.9	+7.2	113.6	135.6	+19.4
	X-2	/	/	/	/	/	/
	X-3	/	/	/	/	/	/
9.4	X-1	47.7	62.3	+30.6	83.8	80.2	−4.3
	X-2	102.1	118.0	+15.6	116.3	120.0	+3.2
	X-3	125.2	105.0	−16.1	151.6	134.0	−11.6

**Table 3 sensors-23-07585-t003:** Effect of mass distribution on the natural frequencies (an example in the X-, Y-, and Z-directions).

Additional Mass (kg)	Distribution Forms	1st-Order Natural Frequency		2nd-Order Natural Frequency	
		Measured Value (Hz)	Deviation (%)	Measured Value (Hz)	Deviation (%)
0	X-1	257.0	0.0	412.0	0.0
	Y-1	257.0	0.0	412.0	0.0
	Z-1	257.0	0.0	412.0	0.0
2.4	X-1	159.3	0.0	236.5	0.0
	Y-1	159.3	0.0	236.5	0.0
	Z-1	159.3	0.0	236.5	0.0
4.2	X-1	177.0	0.0	150.0	0.0
	Y-1	149.3	−15.7	224.2	−4.2
	Z-1	150.0	−15.3	220.0	−6.0
5.9	X-1	134.4	0.0	159.5	0.0
	Y-1	118.4	−11.9	131.3	−17.7
	Z-1	98.0	−27.1	151.5	−5.0
7.6	X-1	81.9	0.0	135.6	0.0
	Y-1	77.3	−5.6	105.9	−21.9
	Z-1	60.0	−2.7	116.2	−14.3
9.4	X-1	62.3	0.0	80.2	0.0
	Y-1	60.5	−2.9	60.3	−24.8
	Z-1	55.3	−11.2	64.1	−20.1

## Data Availability

Data associated with this research are available and can be obtained by contacting the corresponding author.

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
