# Peer review of "Effect of Additional Mass on Natural Frequencies of Weight-Sensing Structures"

_sensors, 2023, doi:10.3390/s23177585_

Round 1

Reviewer 1 Report

Authors have tried to address an essential issue of the Effect of Additional Mass on Natural Frequencies of Weight Sensing structures. The procedure and results presented in the work align with fundamental theories and have been demonstrated using both simulation and practical results.

Some observations are noted below.

- 1st and 2nd harmonic readings deviate significantly for low weights compared to heavier weights. Authors have tried to explain this; however, in a few sentences, can authors also highlight the significance of such deviations for structures having concentrated and uniformly distributed loads? Have authors considered this approach?

- When considering varying loads, what criteria were taken to consider the loads considered in the work?

- Can we identify/design some parameters that will be used to identify loading constraints and their impact on sample structures?

Author Response

Reviewer #1: Authors have tried to address an essential issue of the Effect of Additional Mass on Natural Frequencies of Weight Sensing structures. The procedure and results presented in the work align with fundamental theories and have been demonstrated using both simulation and practical results.

Some observations are noted below.

  • 1st and 2nd harmonic readings deviate significantly for low weights compared to heavier Authors have tried to explain this; however, in a few sentences, can authors also highlight the significance of such deviations for structures having concentrated and uniformly distributed loads? Have authors considered this approach?

- Thanks for the reviewer’s comment. The conclusion for this issue is indeed not rigorous. The issue you mention is important. In different applications, the weight sensing structure may indeed have concentrated loads, uniform loads, and the corresponding laws of influence will be different. In accordance with your comment, we would like to add that the subject of this paper is a type of additional mechanism based on a relatively uniform load distribution (e.g., for high-precision electronic scales with many applications, where the additional mass corresponds to the steel support frame). The mass of additional mechanism is generally not too large (e.g., up to 5 kg) and the load distribution in the X, Y, and Z directions is relatively homogeneous, so that there is hardly any concentration of loads in certain areas. Therefore, the study in this paper should not cover all types of weighing systems in order to avoid lack of rigour. Therefore, the study in this paper should not encompass all the weighing system types to avoid insufficient rigour. In addition, considering the randomness of the load distribution, three different combinations of single-row length extensions, double-rows length extensions and thickness extensions with the same mass are illustrated, to bring out the effect of the different center-of-mass distribution on the natural frequencies, and the related results are also compared with each other, including of the deviations and the corresponding graphs. For applying a large external load at a certain part of the additional mechanism, it can also lead to the occurrence of a sharp increase in the concentrated load. To obtain the law of the influence when applying different loads on the natural frequency, which will be studied as another subsequent topic, we will make further in-depth research in accordance with your comment. A further explanation for this issue has been added in the last paragraph in Part 2.1 in the revised manuscript as:

2.1. Theoretical model

To analyze the laws of influence on the dynamic characteristics of the weight sensing structure with additional mass, a 3-D solid model needs to be established. The 3-D solid model includes a type of additional mechanism based on a relatively uniform load distribution (e.g., for high-precision electronic scales with many applications, where the additional mass corresponds to the steel support frame). The mass of additional mechanism is generally not too large (e.g., up to 5 kg) and the load distribution in the X, Y, and Z directions is relatively homogeneous, so that there is hardly any concentration of loads in certain areas. The additional mass is interconnected with weight sensing structure, which causes the mass to change and causes the natural frequencies to change accordingly. This means that the operational accuracy and stability in practical applications will be limited by the additional mass if no suppression measures are taken. According to the above theoretical model, the additional mass will change the mass distribution of the weight sensing structure, thus affecting the natural frequencies and damping ratio. Therefore, the additional mass is also a cause of resonance in the weight sensing structure.

2) When considering varying loads, what criteria were taken to consider the loads considered in the work?

- Thanks for the reviewer’s comment. The conclusion for this issue is indeed not rigorous. We will follow your comments and add a description of how to choose different loads. The paper for the consideration of different loads, not based on the strict criteria, mainly to consider the coordination of incremental loads. Other considerations are as follows: 1) 2 tons load cell made of aluminum alloy is employed in many applications and manufacturers need to focus on improving the core components, which weighs about 3.50 kg and is a common structural attribute of the decision. 2) The T-shaped loading pads (the 1st additional mass) weighs about 2.40 kg, which is mainly designed according to the basic structure of the maximum load capacity for the 2t. 3) The rest of the additional mass involves a variety of ways to extend the combination, and tends to take half of the load cell mass weighing about 1.75 kg for design, but also to take into account the quality of the more intuitive. In fact, for some high-precision electronic scales of the steel support frame, its load is a random weight, its mass even has a certain degree of randomness, so this paper is a kind of research on the weight sensing structure with the additional mass of the sequential increase in the change rule of the actual natural frequency, including of the influence of other factors. In general, there is no strict criteria for different loads with a particular selection. A further explanation for this issue has been added in the first paragraph in Part 3.2.1 in the revised manuscript as:

3.2.1. Effect of mass distribution on the natural frequencies

The paper for the consideration of different loads, not based on the strict criteria, mainly to consider the coordination of incremental loads. Other considerations are as follows: 1) 2 tons load cell made of aluminium alloy is employed in many applications and manufacturers need to focus on improving the core components, which weighs about 3.50 kg and is a common structural attribute of the decision. 2) The T-shaped loading pads (the 1st additional mass) weighs about 2.40 kg, which is mainly designed according to the basic structure of the maximum load capacity for the 2 tons. 3) The rest of the additional mass involves a variety of ways to extend the combination and tends to take half of the load cell mass weighing about 1.75 kg for design, but also to consider the quality of the more intuitive. In fact, for some high-precision electronic scales of the steel support frame, its load is a random weight, its mass even has a certain degree of randomness, so this paper is a kind of research on the weight sensing structure with the additional mass of the sequential increase in the change rule of the actual natural frequency.

To examine the effect of the additional mass on the natural frequencies of the weight sensing structure, the effect of the additional mass in the X, Y, and Z-directions on the first and second-order natural frequencies of the weight sensing structure is simulated and analyzed, and it is explored whether the combination of the mass distributions causes the natural frequencies of the weight sensing structure to undergo a more significant change, and then experimental validation is carried out, to reveal the effect of the mass distributions on the natural frequencies of the law. Firstly, the influence of X additional mass (single-row length extension X-1, double-rows length extension X-2 and thickness extension X-3) on the first and second-order natural frequencies of the weight sensing structure is studied, and the results of simulation analysis and experimental validation are as follows:

  • Can we identify/design some parameters that will be used to identify loading constraints and their impact on sample structures?

- Thanks for the reviewer’s comment. In fact, in Part 3.2.5 of this paper, some loading constraints have been added during the simulation analysis and experimental validation, such as in Part 3.2.5 of this paper, "The effect of loosening of the fastening screws on the natural frequency of the load cell structure". The loosening of the fastening screws is also a loading constraint, i.e., the maximum torque that can be applied by an adult with each screw tightened is about 150 Nm, whereas in the present experiment, by loosening four screws, the applied torque was reduced to about 30 Nm. The other working conditions were kept unchanged, and then the natural frequency of the load cell structure was affected greatly. In addition, the number of additional masses is also a loading constraint. For example, the number of additional masses is 5, which is based on the basic structure of the most common electronic scales, and the mass of their additional mechanism is generally not too large. The mass of this additional mechanism may be about 1-2 times the mass of the load cell itself, which is of high value for applying research using the most common parameters as loading constraints.

Reviewer 2 Report

1. Authors must include role of Machine learning in sensing structures for weight estimation.

2. How does natural frequency impact weight estimation?

Moderate language editing required

Author Response

Reviewer #2:

  • Authors must include role of Machine learning in sensing structures for weight estimation.

- Thanks for the reviewer’s comment. The proposal on machine learning is a very good research content and an important feature study with great scientific significance. This work will be investigated and applied in feature studies.

  • How does natural frequency impact weight estimation?

- Thanks for the reviewer’s comment. When the natural frequency of the weight sensing structure resonates or resonates with the surrounding external interference sources, it will easily cause the process wiring of the weight sensing structure to be altered or even destroyed, such as weld off, connection loosening or even crack extension. At this time, the output of the weight sensing structure has been seriously deviated from the initial state of the weighing reference value, this is not just a matter of a small error, it may reach more than 50% error, the accuracy of the weighing estimation cannot be mentioned, resulting in the accuracy, reliability and stability of the overall output of the weight sensing structure will not be able to meet the use of the requirements.

  • Moderate language editing required.

- Thanks for the reviewer’s comment. We have further optimized the language editing in several parts of the paper.